# Hot-Wire Laser-Directed Energy Deposition: Process Characteristics and Benefits of Resistive Pre-Heating of the Feedstock Wire

**Agnieszka Kisielewicz** [1,*] (ID), **Karthikeyan Thalavai Pandian** [1] (ID), **Daniel Sthen** [2], **Petter Hagqvist** [3] (ID), **Maria Asuncion Valiente Bermejo** [1] (ID), **Fredrik Sikström** [1] (ID) and **Antonio Ancona** [1,4] (ID)

1   Department of Engineering Science, University West, Gustava Melins Gata 2, 461-32 Trollhättan, Sweden; karthikeyan.thalavai-pandian@hv.se (K.T.P.); asun.valiente@hv.se (M.A.V.B.); fredrik.sikstrom@hv.se (F.S.); antonio.ancona@hv.se (A.A.)
2   GKN Aerospace Sweden AB, Flygmotorvägen 1, 461-38 Trollhättan, Sweden; daniel.sthen@gknaerospace.com
3   Procada AB, Nohabgatan 14, 461-53 Trollhättan, Sweden; petter@procada.se
4   Physics Department, University of Bari, via Amendala 173, 70126 Bari, Italy
*   Correspondence: agnieszka.kisielewicz@hv.se

**Abstract:** This study investigates the influence of resistive pre-heating of the feedstock wire (here called hot-wire) on the stability of laser-directed energy deposition of Duplex stainless steel. Data acquired online during depositions as well as metallographic investigations revealed the process characteristic and its stability window. The online data, such as electrical signals in the pre-heating circuit and images captured from side-view of the process interaction zone gave insight on the metal transfer between the molten wire and the melt pool. The results show that the characteristics of the process, like laser-wire and wire-melt pool interaction, vary depending on the level of the wire pre-heating. In addition, application of two independent energy sources, laser beam and electrical power, allows fine-tuning of the heat input and increases penetration depth, with little influence on the height and width of the beads. This allows for better process stability as well as elimination of lack of fusion defects. Electrical signals measured in the hot-wire circuit indicate the process stability such that the resistive pre-heating can be used for in-process monitoring. The conclusion is that the resistive pre-heating gives additional means for controlling the stability and the heat input of the laser-directed energy deposition.

**Keywords:** laser-directed energy deposition with wire; laser–metal deposition with wire; hot-wire; resistive pre-heating; in-process monitoring





## 1. Introduction

Directed energy deposition (DED) is one category of additive manufacturing (AM) in which materials are melted by focused thermal energy during deposition [1]. In metallic DED, two different types of feedstock material are commonly used: wire and powder [2]. In the case of the applications using wire, the energy supplied to melt the metal is typically generated by either a laser beam, an electric arc or an electron beam. The major advantage of the wire-feed DED processes is material's usage efficiency, which reaches up to 100% [2]. This makes them suitable for production of large-scale components in high volume. Considering this, the wire-feed DED has begun to replace conventional, subtractive machining in various applications where the costs associated with material consumption are considerably high. Hence, the applications of wire-feed DED can be found in the aerospace industry, e.g., mount rings for entire fan cases [3] and features of compressor casings [4] of turbo jet engines as well as space rocket nozzles [5]. Thus, an increased interest in the use of the wire-feed DED for processing of high temperature and corrosion resistant metals, like stainless steels [6–8], Nickel [8–11] and Titanium-based [12,13] alloys, is being observed.

Yet, the wire-feed DED processes share a few common hurdles. To build a complex structure, multiple layers of side-by-side placed beads need to be deposited. While processing, the already deposited metal undergoes multiple heating and cooling cycles, whereas the feedstock wire material is exposed to a cycle of high-rate heating, melting, solidifying and cooling. The amount of heat supplied to the material highly influences its final properties. Thus, the excessive heat input, the high temperature gradients and the heat accumulation related to the multiple heating cycles, can lead to high residual stresses [11], shape distortions [14] and cracking [2] as well as undesirable effects on the resulting microstructure [15–17] and the phase distribution [18]. On the other hand, too low heat input can cause insufficient melting, leading to an insufficient penetration [19] and dilution [8] or lack of fusion (LoF) [19] as well as promote pore formation [20], impairing the mechanical properties [19]. Therefore, the ability to better adjust the heat input is a necessary step for further development of the technology.

Additional drawbacks are limited controllability and repeatability of the processes. Still, a lot of manual adjustment is needed to keep the process stable since no sufficiently reliable monitoring and control solutions are available [21]. From the industrial perspective, this highly restricts the technology to companies that execute both research and production within the field. Hence, for the wire-feed DED systems to be considered turnkey solutions, similar to the powder bed fusion machines, the controllability of the process needs to be improved.

Laser-directed energy deposition with wire (LDEDw), also called laser–metal deposition with wire (LMDw), is one of the wire-feed DED processes. In this technology, a metal wire is melted by a laser beam allowing a flow of the metal from the wire to the melt pool created on a substrate or on a former layer. The stability of LDEDw depends mainly on the character of the metal transfer [22]. The literature describes three different transfer modes: globular, smooth and by plunging [23]. The type of the metal transfer depends highly on the specific energy supplied to the wire and the substrate/previous layer, which is mainly governed by the three process parameters: laser power, wire-feed rate and traverse speed [23]. If the heat input is too high and the wire melts before reaching the melt pool, the globular transfer occurs. In this mode, the forces of surface tension between the molten wire and the melt pool are not strong enough to maintain the continuous link between the materials. Consequently, the wire separates from the melt pool and transfers into the melt pool in form of droplets. This unstable transfer mode results in beads characterized by an irregular shape and internal porosity [23]. If the process parameters are optimized to reach the mode in which the wire melts in a close vicinity to the melt pool, the smooth and stable transfer is established. This is the desired mode as it creates least number of defects and the material with predictable morphology. In the case of insufficient energy supplied to the metal, the wire enters the melt pool in a solid or semi-solid state; the plunging phenomenon, also called stubbing, occurs. In this deposition mode, the solid wire can be in contact with the solid metal below the melt pool causing vibrations of the wire. During plunging LoF defects are created [23]. Hence, to ensure good material quality, a smooth transfer of metal needs to be maintained throughout the entire deposition sequence.

In large-scale production, the main competing technology to LDEDw is wire-arc additive manufacturing (WAAM). In WAAM, an ordinary arc or a so-called plasma arc is used to melt the metal [21]. In comparison to LDEDw, components produced by WAAM, using an ordinary arc, are characterized by the higher heat input as well as the higher probability of process instabilities, defects [24] and shape distortions [21]. Yet, WAAM has two considerable advantages over LDEDw. The first one is the substantially lower cost of the equipment and the maintenance. Indeed, the costs of a high-power laser source, its installation and maintenance as well as the cost of creating $1W$ of laser power, in comparison to the corresponding costs of commercially available arc welding sources, are considerably higher. The second advantage of WAAM is the higher deposition rate in comparison with the LDEDw [25]. Hence, when the high volume and high throughput production is considered, the two mentioned advantages make WAAM the more favorable option.

To improve the competitiveness of LDEDw compared to WAAM technology, resistive pre-heating of the feedstock wire, so-called hot-wire, can be applied. The resistive pre-heating typically occurs while the wire is fed into the melt pool. The energy to pre-heat the wire is generated by an additional electrical power source (EPS). Until now, there is limited information available on applications of the hot-wire in context of LDEDw. Still, the phenomenon has already been applied for arc and laser-based technologies. A common outcome of the pre-heating is the possibility to increase the deposition rate [26,27] or welding efficiency [28]. Additionally, the work performed on Hot-Wire-Arc Additive Manufacturing (HWAAM) indicated that the pre-heating of the feedstock wire contributed to significant reduction in porosity, by limiting pore nucleation sites and creating more favorable conditions for the pores to escape [29]. However, if wire used for deposition was contaminated by greasy dirt and dust particles, the percentage of porosity content increased roughly 30 times, despite the hot-wire being applied [13]. In the case of laser beam welding (LBW), adding an electrical source to pre-heat the wire allowed to maintain the advantage of the low heat input, while decreasing the sensitivity of the process to the placement of the wire with respect to the laser beam [28]. In the case of laser cladding, adding hot-wire allowed reduction of the applied laser power while still maintaining a stable process [30]. Yet, adding resistive pre-heating increases complexity of the processes making them more susceptible to disturbances. For hot-wire-arc-based technologies, the presence of two currents, the feedstock wire current and the arc current, caused the processes to be easily disturbed [29,31], due to the strong interactions of the created magnetic fields. For laser-based technologies, the two main hot-wire parameters, voltage and current, needed to be tuned carefully. If too much electrical energy was applied, the extensive heating of the wire caused globular transfer of the metal, arcing from the wire when it was detached from substrate/previous layer and extensive spatter [32,33]. As an example, when the input voltage was set to 6 V or above, any minor process instabilities resulted in the arcing and the melt pool disturbances [28,33,34]. Both effects are undesirable as they lead to spatter and poor surface quality as well as internal defects, e.g., gas entrapment causing pores. Considering the fact that resistive pre-heating of the feedstock wire had considerable effects on both LBW and laser cladding, when introducing the same technique to LDEDw, one can expect significant changes in the process characteristics, resulting in a creation of a new Hot-Wire LDED (HW-LDED) process.

In this work, we explore the possibility of combining electrical energy and laser energy to gain additional means of adjusting the heat input during LDEDw. To this scope, the influence of the resistive pre-heating of the feedstock wire while depositing was studied. In particular, effects of the interaction of the electrical and the laser energy on the process stability and the morphology of the deposited beads were investigated. The comparison was made between cold process (no hot-wire) depositions and depositions with six different hot-wire levels. The investigation included evaluation of voltage and electrical power supplied in the hot-wire circuit, images acquired by a high-speed camera as well as images of cross-sections of the deposited beads. The results gave insight into the characteristics of a new HW-LDED process, where the combination of electrical and laser energy provides improved control of the heat input and the stability of the deposition process, thus improving fusion of the feedstock wire with the substrate metal.

## 2. Materials and Methods

During this investigation, single beads were deposited using LDEDw with six different levels of the hot-wire. During processing, three different monitoring techniques were used. Subsequently, a metallographic investigation was performed to characterize the morphological features of the beads as well as to quantify possible defects. This section describes the experimental setup as well as the procedures used during LDEDw and the subsequent metallographic investigation.

## 2.1. HW-LDED Processing and Monitoring Setups

The experimental setup consisted of a 6-axis robot, a 6 kW fiber laser, a DED tool with an off-axis wire nozzle attached and a laser optics, a wire feeder, an inflatable chamber with an exhaust system, 3 electrical power sources and a Programmable Logic Controller (PLC). Figure 1 shows the experimental setup.

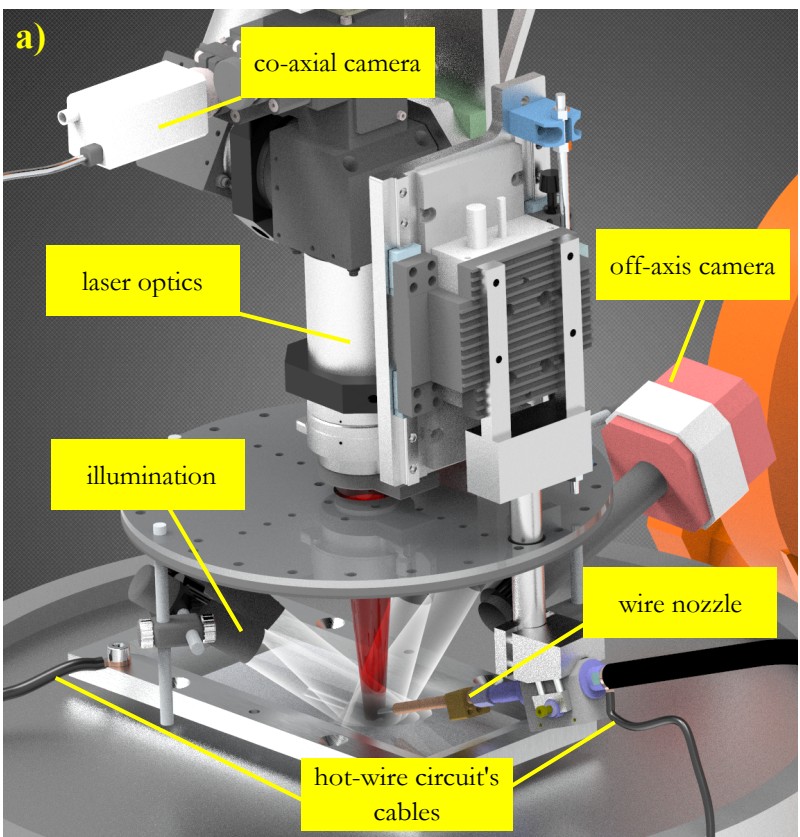

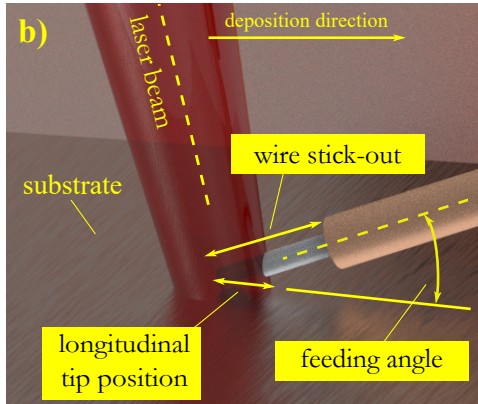

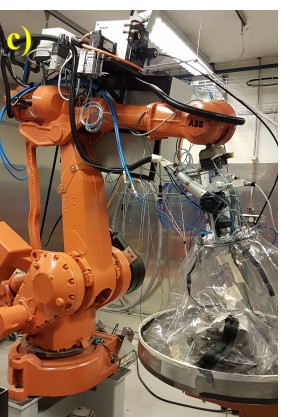

**Figure 1.** The Hot-Wire Laser-Directed Energy Deposition setup: (**a**) the visualization of the deposition tool, (**b**) the schematic drawing of the feedstock wire tip position relative to the laser beam and the substrate, (**c**) the overview of the setup.

The 6 axis IRB-4400 robot (ABB Robotics, Västerås, Sweden) was used to maneuver the DED tool. A copper nozzle guided the wire off-axis into the interaction zone with the laser beam and the melt pool. It also functioned as the electrical contact point, see Figure 2 positive terminal, enabling flow of current through the wire. The setup used the feedstock wire feeder T drive 4 Rob 3 (EWM AG, Mündersbach, Germany) and AM8131 servo-drive (Beckhoff, Verl, Germany). The CX9020 PLC (Beckhoff, Verl, Germany) was used to control

the actuators and to communicate with the industrial robot. The laser beam was generated by the 6 kW YLR-6000-S Ytterbium-doped fiber laser (IPG Photonics, Oxford, MA, USA) working in continuous mode. The wavelength of the beam was 1070 nm. At the focal length of 300 mm, the size of the spot was 1.5 mm in diameter. As the DED wire process requires a larger spot size, the experiments were conducted using the out-of-focus spot size of approximately 3.2 mm in diameter and a Gaussian beam power distribution. To provide shielding from reflected laser light, two cooling plates were attached to the tool. One was placed in the back of the nozzle, protecting the nozzle mounts, electrical cables and tubes. The second one, a round plate mounted directly below the laser optics, protected the optics and the linear actuator. Both plates were water cooled. A fixture, providing mount for a substrate metal plate, served also as the contact point for the hot-wire circuit, see Figure 2 negative terminal.

To provide electrical energy, an EPS system was used consisting of the three electrical power sources EA-PS 8080-60T (EA Elektro-Automatik GmbH & Co KG, Viersen, Germany) connected in parallel. These EPSs were able to supply direct current (DC) of maximum 180 A and a total power of maximum 4500 W. The EPSs worked in a constant voltage mode, where the output voltage $U_{set}$ and the load resistance determined the current. The defined response time to load changes for single EPS was <2 ms. The voltage set-point was given as a signal by the PLC using an analogue 0–10 V interface. To compensate for the voltage drop over the power cables, the controlled voltage was measured by the EPSs directly across the process load.

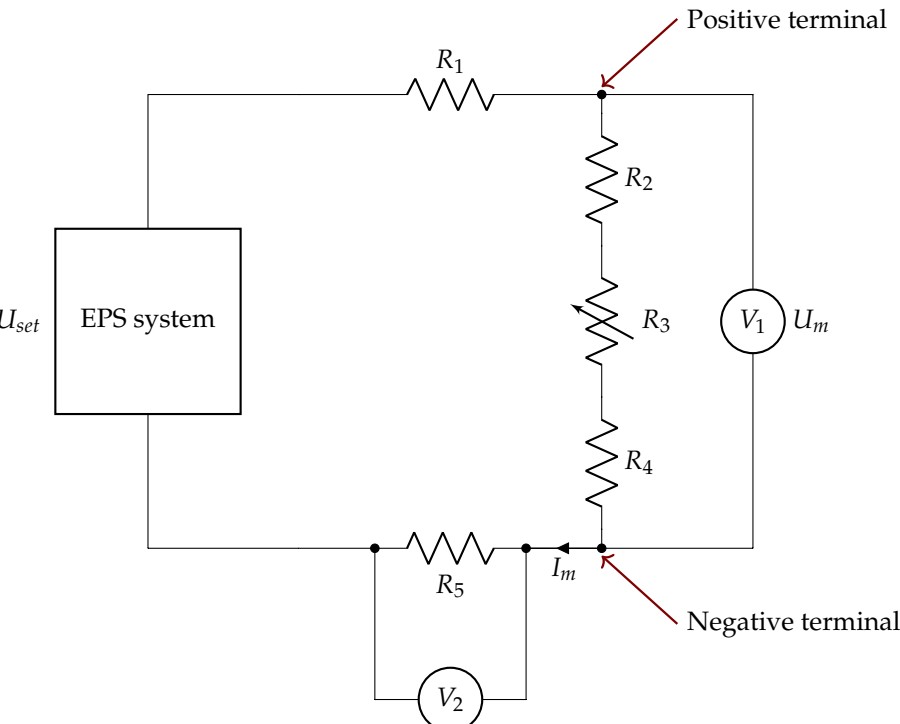

**Figure 2.** The circuit diagram of the hot-wire system where $R_1$ is the resistance of the power cables, $R_2$ is the resistance of the nozzle, $R_3$ is the resistance of the feedstock wire and the melt pool (process load), $R_4$ is the resistance of the fixture, $R_5$ is the resistance of the current shunt, $V_1$ is the voltage drop between the nozzle and the fixture including the process load and $V_2$ is the voltage drop over the current measuring shunt.

The monitoring system consisted of three different subsystems: a voltage and current measurement circuit, a vision camera integrated co-axially with the laser beam and a high-speed vision camera mounted off-axis, see Figure 1a. The electrical signals were measured using two PLC modules, EL3751 and EL3314 (Beckhoff, Verl, Germany), respectively. The modules have the accuracy of ±0.01% of voltage measurement and <0.3% of current

measurement. The voltage drop between the nozzle and the fixture, including the process load, was measured directly, see *V*1 in Figure 2. Compared to the resistance of the process load *R*3, the values of the nozzle resistance *R*2 and the fixture resistance *R*4 were negligibly small. The current in the circuit was indirectly derived from the voltage drop over the 1 6053-20 current shunt (Thermovolt AB, Hallstahammar, Sweden), see *V*2 in Figure 2. The shunt was connected in series with the process load, see *R*5 in Figure 2. The sampling rate of the signals was 30 Hz. The electrical data were logged to a database. A circuit diagram of the hot-wire system is presented in Figure 2.

The co-axially integrated GigE acA1600-20gm camera (Basler AG, Ahrensburg, Germany) allowed real-time observations of the laser–metal interaction zone. The scene was continuously illuminated using blue color light emitting diode (LED) lamp M450D3 (Thorlabs Inc., Town of Newton, NJ, USA) producing light with peak intensity at the wavelength of 450 nm. A hard-coated bandpass filter transmitting light in the same wavelength range was used in the optical path of the camera. This was done to facilitated reduction of the light interference from the vapor plume and saturation of images caused by the bright laser light reflected from the melt pool. The camera was triggered as the laser started to operate. The exposure time was 33.2 ms and the frames were acquired with the frequency of approximately 30 fps. Additionally, four blue color M450D3 LED lamps (Thorlabs Inc., Town of Newton, NJ, USA) were mounted to provide a bright field for the image acquisition of a high-speed complementary metal oxide semiconductor (CMOS) CCM-1540 camera (Integrated Design Tools Inc., Pasadena, CA, USA). The camera was mounted off-axis, with a low angle relative to the substrate surface, to capture the interaction of the wire with the melt pool from the side. As in case of the co-axial camera, a hard-coated band pass filter was used. Images were acquired by software Motion Studio x64 (Integrated Design Tools Inc., Pasadena, CA, USA) with a frame rate of 800 fps. The exposure time was set to 200 µs. To obtain in-depth information on the wire-melt pool interaction, the analysis of images captured by the off-axis camera was performed off-line.

The Deposition of Beads

During the experiments, single bead samples were produced. The process parameters are given in Table 1. The front feeding direction was used. The longitudinal wire tip position was adjusted so that the wire was in contact with the substrate in the center of the laser beam. Figure 1b shows the details of the wire's positioning relative to the substrate and the laser beam. To limit problems related to arcing while depositing, the voltage levels ($U_{set}$) used were restricted to the range between 0.5 V and 3 V, see Table 2. For each combination of the hot-wire level (L0–L6) and the laser power level (low, medium, high), three beads were deposited. The sequence of the depositions was randomized to reduce the effects of localized heat build-up in the substrate metal and its deformation. The additive material used to deposit the beads was a 1.2 mm in diameter Duplex stainless steel 2209 wire. The beads were deposited on $300 \times 100 \times 10$ mm Duplex stainless steel 2205 substrates. The measured chemical compositions of a substrate and a wire of the same batch as the processed material are shown in Table 3. Before processing, the substrates were cleaned. In the first step, the top surface of a substrate was manually polished with steel wool. Afterwards, the substrate was chemically cleaned with acetone. While processing, the deposition tool was tilted 10° from the normal direction to the substrate's surface to avoid back reflections of the laser beam into the optics.

**Table 1.** Processing parameters.

| Laser Power | | Wire-Feed Rate | Traverse Speed | Stick-Out | Feeding Angle |
|---|---|---|---|---|---|
| [-] | [W] | [m/min] | [mm/s] | [mm] | [deg] |
| low | 2500 | 2 | 10 | 6.5 | 18.6 |
| medium | 3500 | 2 | 10 | 6.5 | 18.6 |
| high | 4500 | 2 | 10 | 6.5 | 18.6 |

**Table 2.** Selected levels of the set voltage $U_{set}$ and corresponding hot-wire levels.

| Hot-Wire Level | [-] | L0 | L1 | L2 | L3 | L4 | L5 | L6 |
|---|---|---|---|---|---|---|---|---|
| $U_{set}$ | [V] | 0 | 0.5 | 1.0 | 1.5 | 2.0 | 2.5 | 3.0 |

**Table 3.** Chemical composition (wt%) [35].

| Material | C | Si | Mn | Ni | Cr | Mo | N | S | P | Cu | Co |
|---|---|---|---|---|---|---|---|---|---|---|---|
| Substrate | 0.016 | 0.32 | 1.77 | 5.50 | 22.77 | 3.07 | 0.177 | <0.001 | 0.027 | 0.21 | 0.096 |
| Wire | 0.016 | 0.45 | 1.45 | 8.62 | 23.23 | 3.29 | 0.160 | 0.001 | 0.016 | 0.04 | - |

*2.2. Metallographic Investigation*

As this study focuses on the stability of the process and interaction effects of the electrical and the laser power, the included metallographic investigation is restricted to the morphology of the beads and the defect evaluation. An extended metallurgical investigation on Duplex stainless steel processed with HW-LDED can be found in [35].

To prepare specimens, a conventional metallographic preparation procedure was followed, and cross-sections were polished up to 0.05 µm. Subsequently, the etching procedure was performed on the specimens. The specimens were etched with freshly prepared modified Beraha etchant consisting of 66% water, 33% hydrogen chloride and 1% potassium bisulfite. The etchant was active on the specimen surface from 10 to 12 s.

The prepared specimens were examined in an Axio Imager. M2m light optical microscope manufactured (Carl Zeiss AG, Jena, Germany) to investigate bead morphology and to detect presence of internal defects. The geometric features, like height, width, penetration depth and dilution, were investigated. Using the inbuilt software in the microscope, multiple images of different regions of the cross-sections were taken at the ten-fold magnification. Afterwards, the images were combined to visualize the entire cross-section. Figure 3a presents an example of a cross-section. The dilution ratio ($\eta$) was calculated by the following equation

$$\eta = \frac{B}{A + B} \times 100\% \tag{1}$$

where *B* is the penetration area and *A* is the added material as shown in Figure 3b.

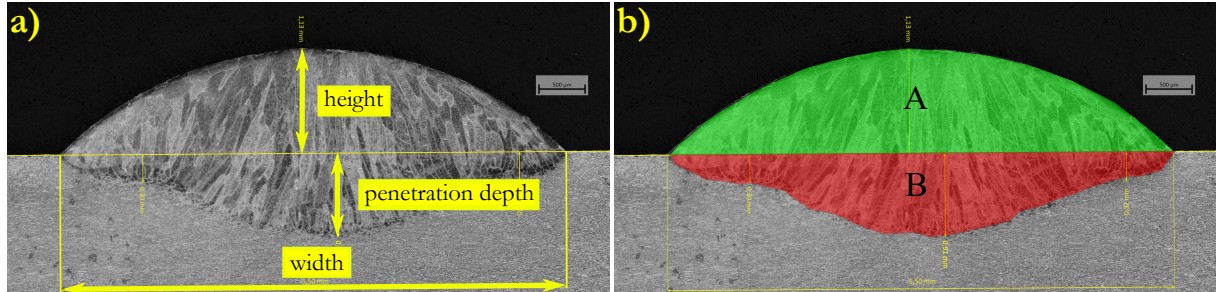

**Figure 3.** The features of the single bead geometry: (**a**) height, width and penetration depth, (**b**) added material (A) and penetration area (B).

*2.3. Statistical Analysis*

To characterize the influence of the resistive pre-heating and the laser power on the geometric features, the full factorial statistical analysis was applied. The hot-wire level and laser power level were selected as independent variables. Height, width, penetration depth and dilution were considered to be the responses. For each response, three values, each measured on a cross-section obtained from a separate deposition, were analyzed. The analysis was performed with 95% confidence level, corresponding to $\alpha = 0.05$. The analysis included hot-wire levels between L0 and L3 where for most of the depositions, the process

was stable. The remaining data related to unstable process conditions were excluded from the statistical analysis as they provide little usefulness when setting up real-case depositions. The analysis was performed using Minitab software, using the inbuilt DOE toolbox. To judge which factors and their interaction had a statistically significant influence on each of the responses, Pareto charts of standardized effects were created and analyzed.

## 3. Results and Discussion

To understand the influence of the hot-wire on the LDEDw process, the monitoring data and the data obtained from the metallographic investigation were analyzed. In this section, the data as well as the analysis are presented and discussed.

### 3.1. Hot-Wire: Modes

Based on the acquired online electrical signals the analysis of the resistive pre-heating was conducted in order to separate stable process modes from unstable one. To judge stability of the process, levels of measured average voltage $U_m$ as well as its standard deviation were analyzed. For understanding the magnitude of the phenomenon, average electrical power $P_{el}$ levels were compared as well as their standard deviations.

#### 3.1.1. Stable Mode

Figure 4 shows mean values of the measured voltage $U_m$ during each deposition, to the left, and the standard deviations of $U_m$, to the right. Figure 4a,b show the voltage and the standard deviation for low laser power. When the voltage $U_{set}$ was in the range between 0 and 1.5 V, i.e., hot-wire levels L0 to L3, the measured voltage values $U_m$ were in a good correspondence to the $U_{set}$-values. The standard deviation values were on a comparable level and did not exceed 0.05 V. Similar behavior was observed for the depositions performed with medium laser power, see Figure 4c,d. In the case of high laser power, the strong correspondence between the set $U_{set}$ and the measured voltage $U_m$ as well as the low standard deviation was observed only for the depositions performed with the $U_{set}$ in the range between 0 and 1.0 V, see Figure 4e,f.

Figure 5 presents average values of the electrical power during depositing with the three different levels of laser power. The values were determined based on the measured voltage $U_m$ and current $I_m$ values, calculating a product of a multiplication of each pair of the values, afterwards deriving average of the products for each single deposition. For the experiments performed with the low and medium laser power, when the hot-wire level was between L0 and L3, the average electrical power increased linearly with the increase of the level of the hot-wire. Here, the average values of electrical power were comparable among all three depositions and the maximum value was below 200 W, see Figure 5a,b. During processing with high laser power, a similar trend was observed for samples deposited with hot-wire level between L0 and L2. In these cases, the maximum obtained power was below 75 W.

For the described above conditions, the process of pre-heating of the feedstock wire had stable characteristics, giving repeatable and predictable results relative to the electrical power.

#### 3.1.2. Unstable Mode

When the set voltage $U_{set}$ increased above 1.5 V for the low and medium laser power and above 1.0 V for the high laser power, the characteristics of the measured voltage $U_m$ changed. The values of $U_m$ were not anymore corresponding to the set values of $U_{set}$, reaching on average higher levels, see Figure 4a,c,e. Furthermore, the variation among data samples of measured voltage during each deposition sequence increased as well, reaching values between 0.2 V and 1.07 V, see Figure 4b,d,f. Similar behavior of increased standard deviation was observed in the current signals.

In addition a larger variation between the average electrical power values of samples deposited in the same process conditions was observed, see Figure 5. For each deposition, the

standard deviation of the electrical power increased as well. This was expected as for these process conditions the voltage $U_m$ and current $I_m$ signals showed much more variation.

Hence, the observed changes of characteristics of both measured voltage $U_m$ and electrical power indicated deviation from the stable mode.

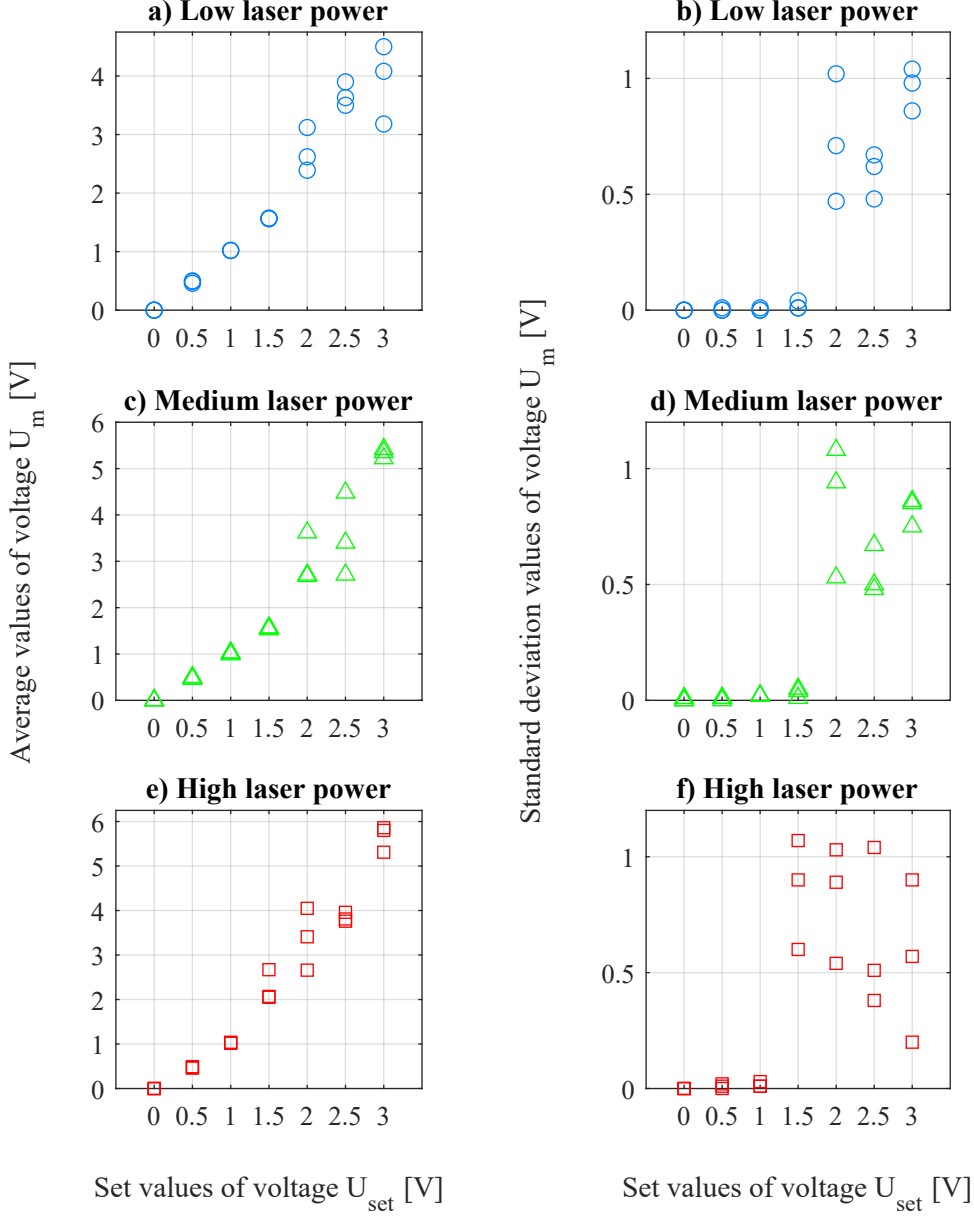

**Figure 4.** The average values of the measured voltage $U_m$ and the standard deviations of each deposition in relation to the set voltage $U_{set}$: (**a**,**b**) low laser power, (**c**,textbfd) medium laser power, (**e**,**f**) high laser power.

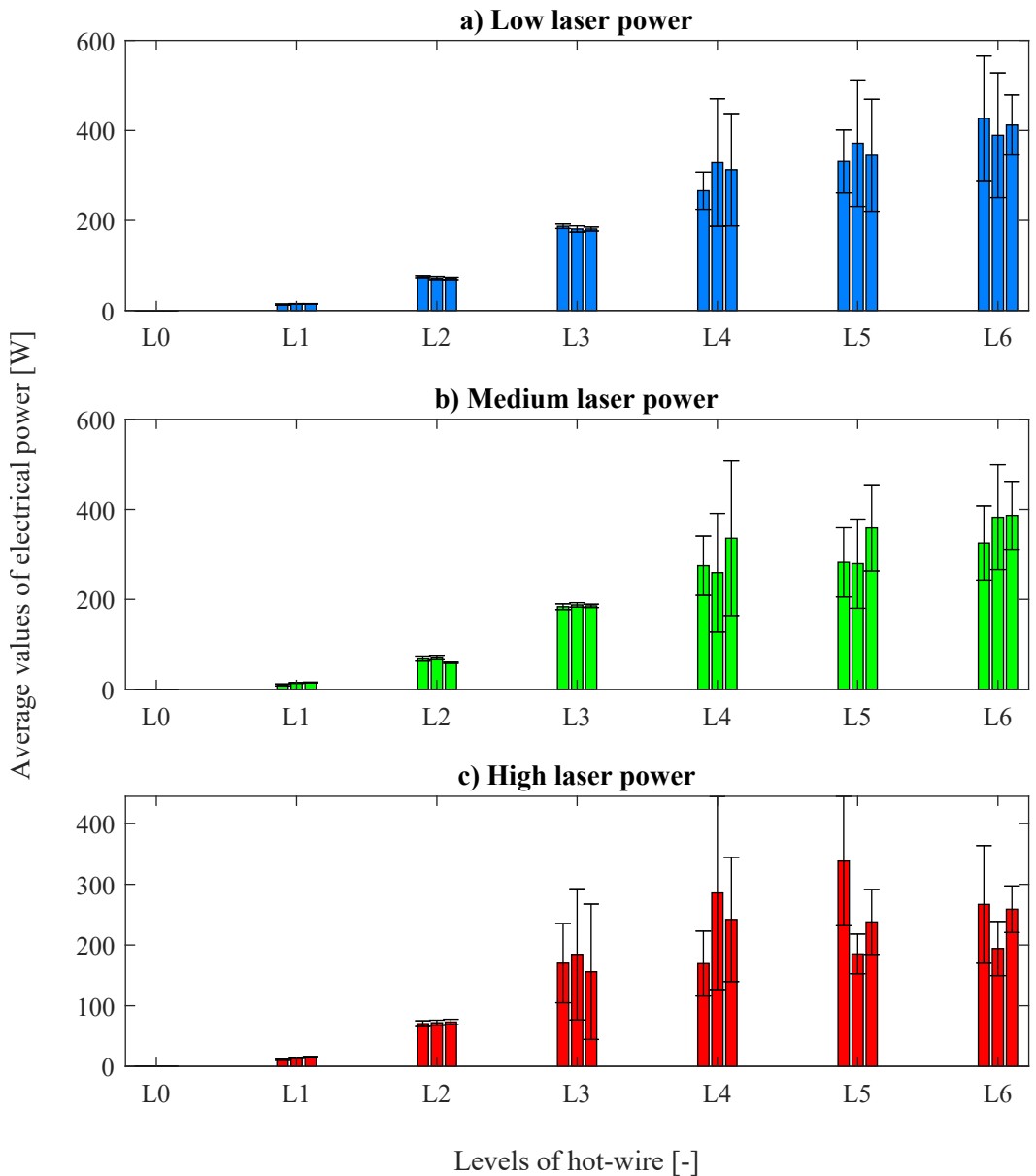

**Figure 5.** The average electrical power relative to each hot-wire level: (**a**) low laser power, (**b**) medium laser power, (**c**) high laser power.

### 3.2. Monitoring of the Metal Transfer

The images captured by the high-speed camera gave an additional insight into the characteristics of the metal transfer relative to the hot-wire level. Figure 6 presents images acquired at 1.08 s, approximately 10 millimeters from the start, of the three depositions characterized by the different levels of the hot-wire with the medium laser power. For the low hot-wire level (L1), see Figure 6a, the wire, being partially solidified, entered much deeper into the melt pool than in the other two cases. The images revealed that as the wire traveled through the laser beam, the metal located on the top surface of the wire gradually melted creating a concave profile, see marked green contour in Figure 6a. The bottom surface of the wire, closer to the substrate, remained solid, shadowing the substrate metal from the laser light. The melt pool was characterized by a concavity located in the front edge of its contour. In that area, there was a possibility of solid-to-solid metal interaction as it was not clear at which point the wire reached the fully melted state. Yet no clear indications of stubbing were observed. For the medium level (L3), see Figure 6b, the images indicate that the wire did not enter the melt pool in the partially solidified state as deep

as in the previous case. Thus, the substrate was less shadowed by the wire and the area of the concave profile on the top surface of the wire was less evident. In the case of the high hot-wire level (L6), see Figure 6c, the wire changed its state from solid to liquid much sooner than in the two previous cases. This be can observed based on the appearance of the wire tip and the extent of the melt pool. No concave profile was visible on the top surface of the wire, see Figure 6c. As a considerable amount of the wire became melted simultaneously, the gravitational force was high enough to overcome the liquid surface tension, creating a narrower transition link. As long as the connection between the wire and melt pool was maintained, the melt pool shape was elongated in the same direction as the wire. Under such process conditions, the transfer of metal was unstable.

Figure 7 presents a sequence of images revealing the unstable material transfer. The decrease of the cross-section (see Figure 7a,b) led to the break of the transition link between the molten wire and the melt pool. This was often accompanied by an electrical discharge, see the increase of bright light in Figure 7c. Additionally, spatter was observed, see Figure 7d. Pressure created during the discharge pushed the melted material further away from the wire inducing waves on the surface of the melt pool, see Figure 7e,f. As the wire continued to travel through the laser beam a small droplet started to form on the wire tip, see Figure 7f. When the gravitational forces surmounted the surface tension of the molten tip of the wire, the metal dropped abruptly into the melt, temporally re-engaging the link. This phenomenon occurred frequently during the depositions for all laser power levels when hot-wire was higher than L3.

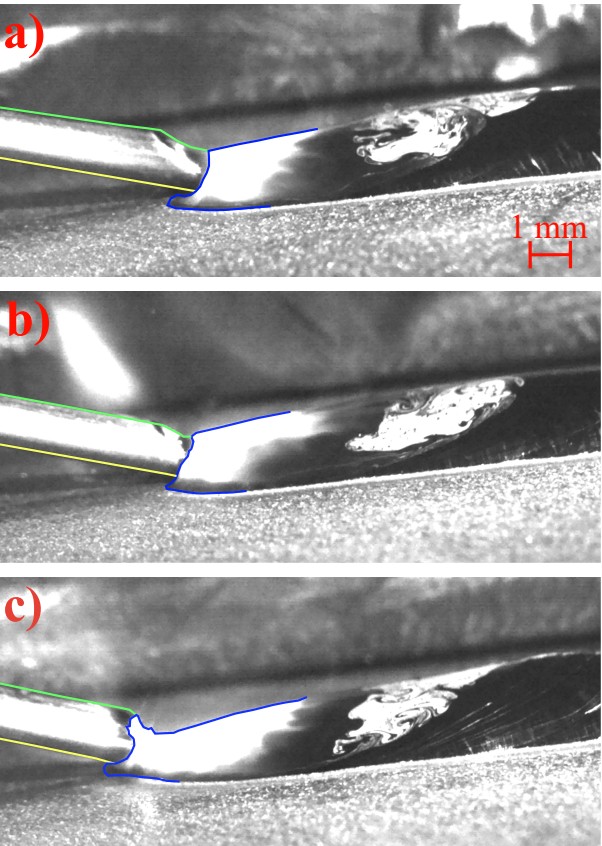

**Figure 6.** The wire-melt pool interaction acquired by the off-axis high-speed camera when three different levels of hot-wire and the medium laser power were applied: (**a**) L1—low hot-wire level, (**b**) L3—medium hot-wire level, (**c**) L6—high hot-wire level.

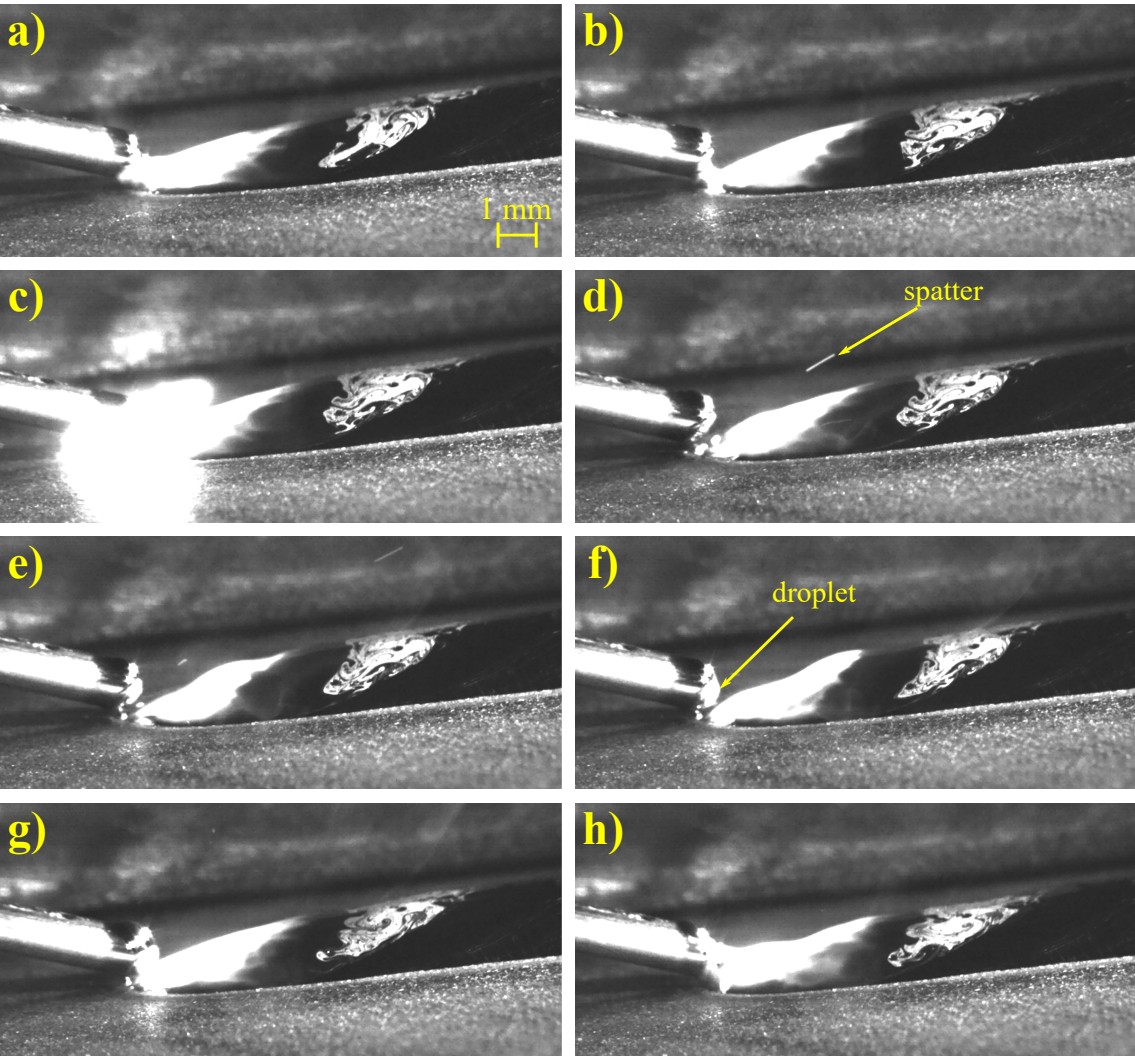

**Figure 7.** The sequence of images acquired by the off-axis high-speed camera showing the process instability when the medium laser power and high hot-wire level (L6) were applied: (**a**) the transition link between the molten wire and the melt pool ($t$ = 1.659232 s), (**b**) the narrowing transition link between the molten wire and the melt pool ($t$ = 1.682776 s), (**c**) the arcing occurring with the break of the transition link ($t$ = 1.684015 s), (**d**) the spatter ($t$ = 1.685254 s), (**e**) the disturbed melt pool surface ($t$ = 1.686493 s), (**f**) the start of droplet formation ($t$ = 1.686493 s), (**g**) the development of the droplet ($t$ = 1.697646 s), (**h**) the temporarily reconnection of the molten wire and the melt pool ($t$ = 1.702602 s).

### 3.3. The Influence of the Hot-Wire on the Deposition Stability

From the electrical signals and the camera images, different process behaviors were observed depending on the amount of the supplied energy and the type of the energy source. Hence, increasing the laser power, when no hot-wire was used, did not significantly influence the characteristics of the material transfer. For all three laser power levels it remained stable. However, using the resistive pre-heating decreased the required time for the wire to melt while being heated by the laser beam; finally leading to the transition from the smooth to the globular material transfer. This can be explained by the following physical, non-linear mechanism: as the current was flowing through the wire, it pre-heated the material. With the increase of the level of the hot-wire, more energy was directly supplied to the wire and to the melt pool. As the wire temperature increased, its reflectivity at the laser wavelength decreased, allowing more laser light to be absorbed by the wire [36,37]. Therefore, the total heat input increased. In the case of the highest hot-wire levels, the metal reached the melting temperature before reaching the melt pool and the metal transfer changed from smooth to globular. The threshold at which the transition from

the stable to the unstable transfer occurred depended on the interplay between the laser power and the hot-wire level. For maintaining stability at the low and medium laser power, it was possible to supply up to approximately 188 W of electrical power, corresponding to the hot-wire level L3. Whereas, when the high laser power was applied, the maximum electrical power supplied was approximately 73 W, corresponding to the hot-wire level L2.

Unlike in the case of the hot-wire-arc-based processes [31], no negative coupling between the resistive pre-heating of the feedstock wire and the laser beam was observed, as they are governed by different physical phenomena. The observed instabilities of the process, i.e., droplets, were related to excessive heat input into the wire. Thus, when adjusting levels of the laser power and the hot-wire, one can focus mainly on the required heat input and final bead geometry.

Considering all, the hot-wire provides means of influencing the characteristics of the deposition process. Well adjusted, it improves the metal transfer, allowing the laser energy to be used more efficiently. Moreover, a globular transfer indicates too high energy input. This can give an opportunity to extend the process window by increasing the wire-feed rate and the deposition speed, simultaneously increase the throughput of the process, similar to laser cladding [38]. Another alternative could be to decrease the laser power and by this enable applications with low power laser sources. In this case, the initial investment cost of the LDEDw technology would decrease, making it a more affordable solution.

It must be stated that the values of the electrical power explored here can only serve as an indication of the limits of levels of the hot-wire since establishing the exact threshold was not the aim of this investigation.

It is worth noting that similarly to the LBW [39] and the laser cladding [38], the hot-wire processes, the voltage and the current not only influence the process window but also enable in-process monitoring. During the experiments, when the deposition process was stable and the metal transfer was smooth, the measured levels of the voltage $U_m$ did correspond to the set levels $U_{set}$ and the standard deviations for both voltage and current were low. The camera images confirmed that the transition link between the wire and the melt pool was maintained throughout the entire deposition sequences allowing the hot-wire current to flow through the metal without interruptions.

As the electrical power increased above a certain threshold, the process became unstable, and the metal transfer changed from smooth to globular. This change was easily observable by the increased standard deviation of the electrical signals. As the link between the molten wire and the melt pool was becoming narrower, the resistance of the metal was increasing and less current could flow through the metal. At the moment of the break of the transition link, the resistance increased abruptly, which is clearly indicated by the measured voltage signal. Thus, the electrical signals have the potential of providing a reliable means of monitoring and controlling the quality of the metal link between the melting wire and the melt pool.

### 3.4. The Influence of the Hot-Wire on the Deposited Beads' Characteristics

After deposition, the as-built beads were subjected to the visual inspection as well as the subsequent characterization of the following geometric features: height, width, dilution and penetration depth, as defined in Figure 3 and Equation (1). The features were evaluated based on the cross-sectional images, see examples in Figure 8. Further statistical analysis of the influence of both the laser power and the hot-wire level on the bead's geometric features was performed. Additionally, presence of surface as well as internal defects was investigated.

### 3.4.1. Height

The average values and the standard deviations of the height are presented in Figure 9a. Compared to the average height measurements of the samples produced with cold wire, the addition of the hot-wire produced a change of the height varying between $-7.4\%$ and $18.9\%$. For the beads produced with the hot-wire level between L2 and L5, with the increase

of the resistive pre-heating, an increase of the height was observed. One can also observe an increase in the standard deviation of the height for the specimens produced by the unstable process. This can be explained by the uneven surface of the samples, described in detail in Section 3.4.5.

Figure 10a presents the Pareto chart of the standardized effects showing the influence of the two factors. The results of the statistical analysis indicate that the laser power has the largest influence on the height. An increase in the laser power results in a decrease in the height. Although the hot-wire proves to be statistically significant, its impact is much lower. The interaction between the two factors is below the threshold of statistical relevance. Thus, it can be disregarded. Considering all, if the bead's height requires adjustments, it can be better achieved by tuning the laser power.

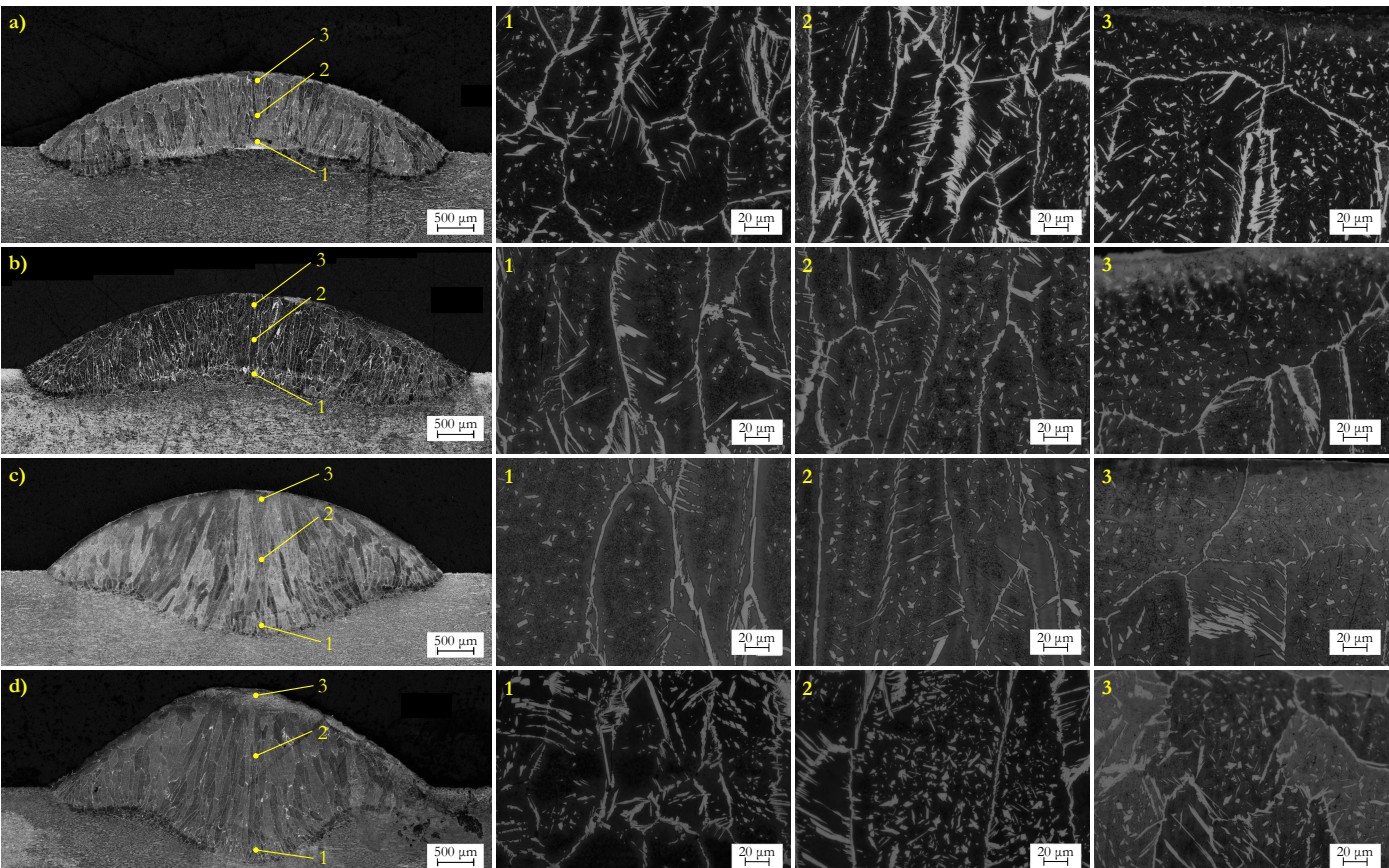

**Figure 8.** The cross-sections of the beads produced with the medium laser power and different levels of hot-wire: (**a**) L0—cold wire, (**b**) L1—low hot-wire level, (**c**) L3—medium hot-wire level, (**d**) L6—high hot-wire level.

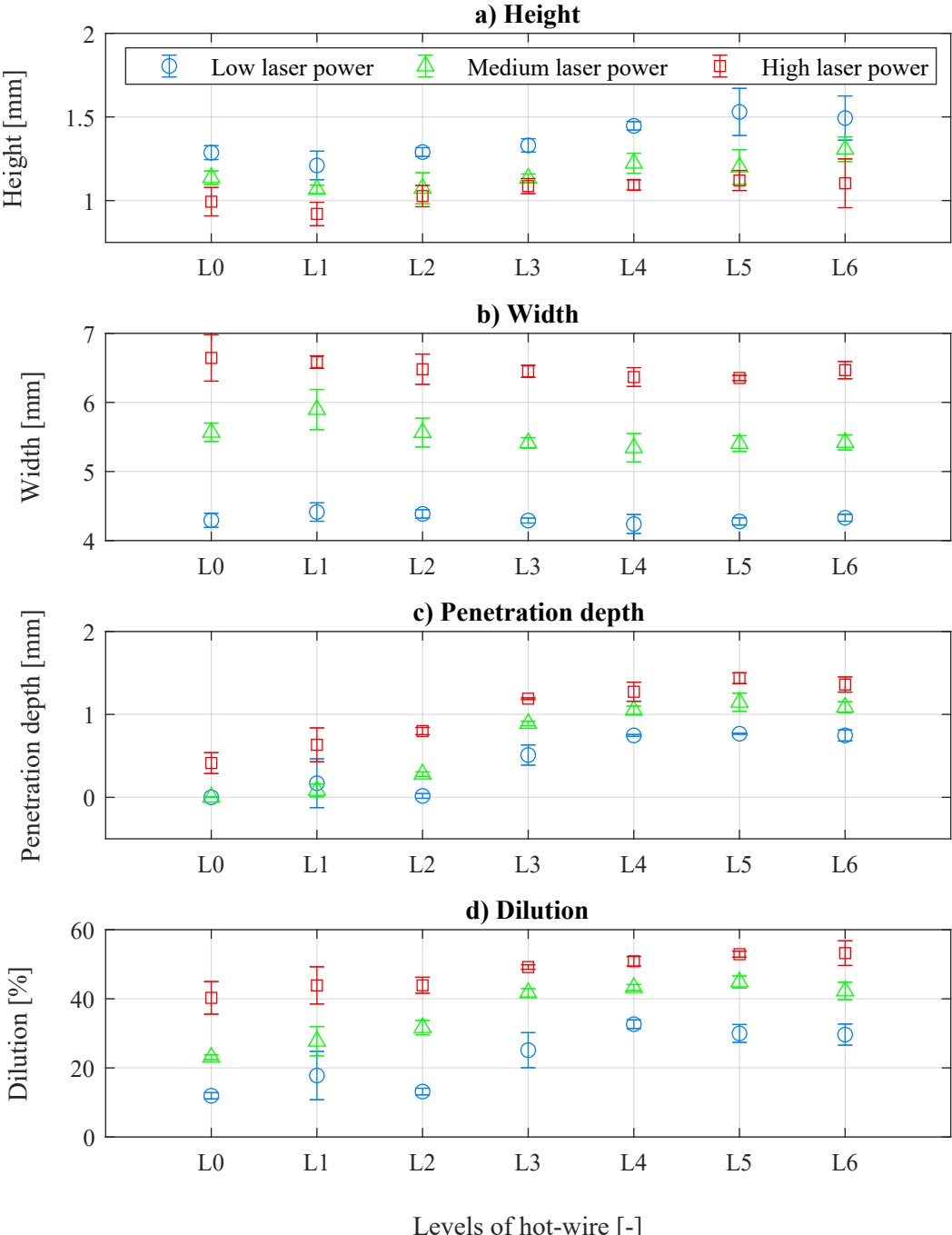

**Figure 9.** The average values of measured geometrical features of the beads: (**a**) height, (**b**) width, (**c**) penetration depth, (**d**) dilution.

3.4.2. Width

Figure 9b shows how the width is influenced by the hot-wire levels and laser power. Overall, the width of the hot process beads varied between −4.3% and 5.9% with respect to the cold process. The addition of the hot-wire did not have a significant impact on the measured width. On the contrary, varying laser power resulted in considerable changes of the width. Thus, increasing the laser power resulted in increasing the width, which is in correspondence with findings presented in [23].

The significance of the laser power influence was also confirmed by the results of the statistical analysis, see Figure 10b, where the laser power is the most dominant factor and the hot-wire barely exceeds the threshold of statistical significance. It was also observed that for majority of specimens, the variations in the width were inversely correlated with

the variations in the height. This was due to the fact that the volume of deposited material was constant during all depositions. If more material remained in the middle of the bead, increasing height, less material could flow to the sides of the beads. Thus, to adjust the width, the laser power should be tuned. However, it is necessary to regard the coupling between the width and the height, as a change in one of them will influence the other.

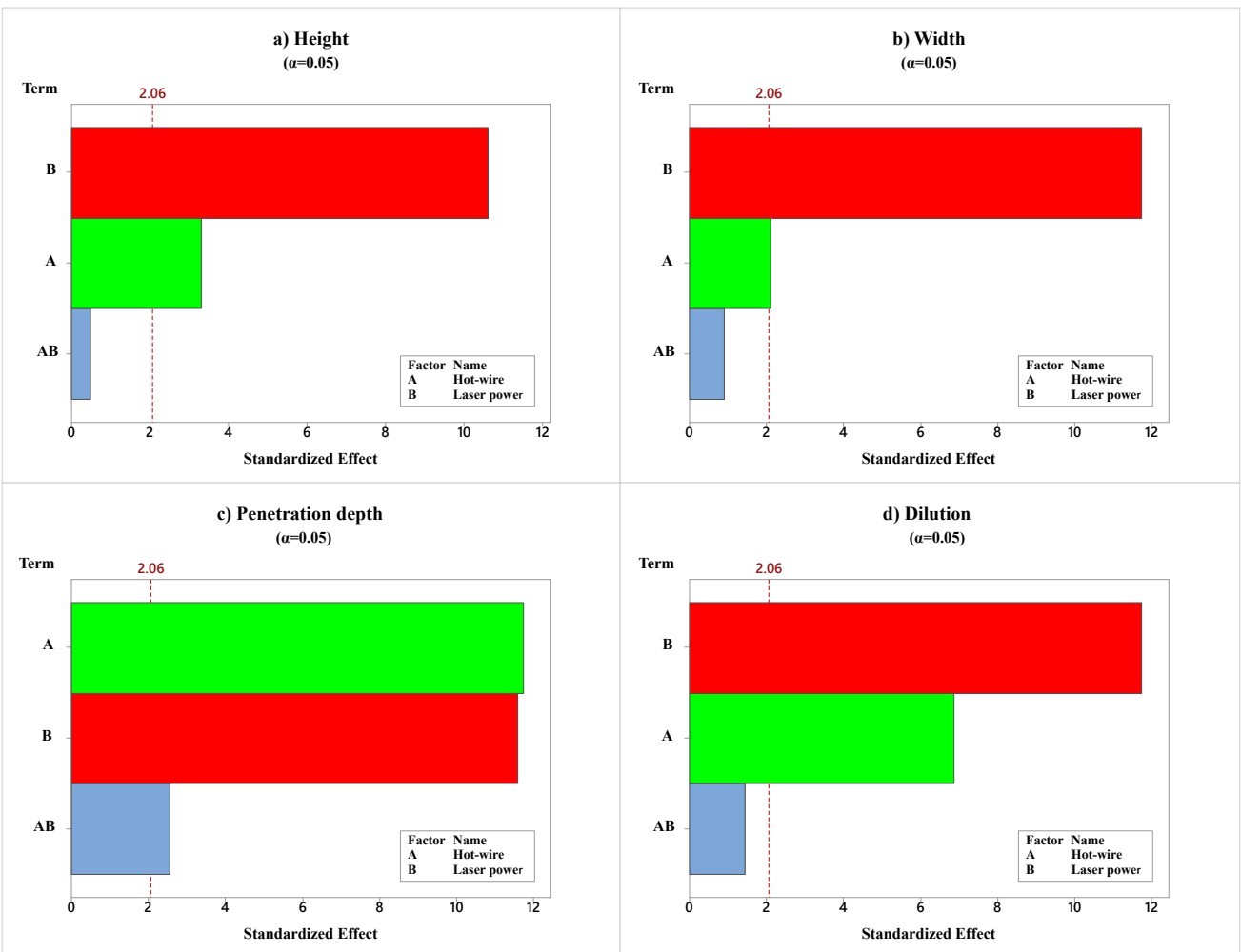

**Figure 10.** The standardized effects of the laser power and the hot-wire (level L0-L3) on the geometric features: (**a**) height, (**b**) width, (**c**) penetration depth, (**d**) dilution. The threshold value of statistical relevance for this analyzes was equal to 2.06 according to the 95% confidence level and the t-student distribution.

### 3.4.3. Penetration Depth

Figure 9c shows the evolution of the penetration depth relative to the increase of the hot-wire level. At the low laser power, the beads deposited with cold process were characterized by a limited penetration or lack of penetration in the fusion area close to the toes of the beads. The average penetration depth increased to 0.51 mm as the hot-wire was set to level L3. An increment of the hot-wire by one level increased further the penetration, which reached on average 0.75 mm. Further increase of the hot-wire level did not bring any major change in the penetration depth. Similar relations were observed for the beads produced with the medium laser power. In the cold process conditions, no penetration as well as fusion were observed in the middle of the bead, as it is shown in Figure 8a. A considerable increase of penetration depth was achieved when the hot-wire was set to level L3. In this case, the average measured penetration depth was equal to 0.89 mm. Further increase of the hot-wire level caused increase of the extent of the penetration, which

reached on average 1.09 mm for the beads deposited in the unstable process conditions. For the high laser power, the energy supplied by the laser beam was enough to melt both the wire and the substrate, without the necessity of adding the resistive pre-heating. The cold process penetration depth was, on average, 0.41 mm. As with the low and medium laser power cases, the addition of the hot-wire caused an increase in the penetration depth. Within the stable process window (L0–L2), the highest average penetration depth was equal to 0.8 mm. Further increase of the hot-wire level caused instability and even deeper penetration, which reached the highest average value of 1.44 mm for the hot-wire level L5.

It is worth mention that the penetration profile was not uniform. It increased gradually from the sides of the bead, experiencing large gradient of increase in the middle section. The phenomenon became more prominent when the hot-wire levels exceeded L3 and the wire became prematurely melted causing droplets, see Figure 8d.

The overall observed trend was that the penetration depth increased with the addition of the hot-wire. The extent of the influence depended on the laser power level used during processing. As an example, adding on average 186 W of electrical power in combination with the medium laser power gave good quality beads and penetration depth of 0.89 mm. Whereas, when increasing only the laser power, it was necessary to increment from medium to high level, by adding 1000 W, to achieve the penetration depth of 0.41 mm. Thus, the electrical power is more effective, in terms of improving the penetration depth, than the laser power as less power is needed to reach comparable penetration depths. This can be related to the fact that the efficiency of heating the metal by electrical current is close to 100% as the energy is directly supplied to the wire. In addition to that the resistive pre-heating of the wire improves absorptivity of the laser light, as discussed earlier in Section 3.3, allowing the combined heat input to be higher.

The results of statistical analysis confirm that both the laser power and the hot-wire, are relevant for the adjustments of the penetration depth, see Figure 10c. They are characterized by comparable values of the standardized effect, see Figure 10c. Furthermore, the interaction between the two factors is also statistically significant. Hence, the penetration depth can be influenced in three different ways, i.e., by increasing the laser power or the hot-wire level separately or by manipulating the two parameters. Considering that the hot-wire does not impact the height and width to the same extent as the laser power does, adjusting the resistive pre-heating can be the preferable way to achieve improved fusion while maintaining width and height at the similar levels.

### 3.4.4. Dilution

The effect of the hot-wire on the dilution is presented in Figure 9d. The dilution of the beads deposited with the high laser power experienced the least dependence. For the hot-wire levels L1 and L2, the dilution increased by 9.0% compared to cold process, reaching on average the level of 43.9%. The highest level of the dilution was reached for the hot-wire level L6 and was equal to 53.2%. For the medium laser power, the extent of the dilution almost doubled compared to the cold process, reaching up to 41.7% for the hot-wire level L3. The further increment of hot-wire level, did not bring any significant increase in the dilution. In the case of the low laser power, the dilution increased on average from 11.9 to 25.2% when the hot-wire increased from level L0 to L3. This means more than two-fold increase compared to the cold process conditions. The further increase in the hot-wire level contributed to an increase of the dilution as the value reached at the highest 32.6% for the hot-wire level L4, increasing almost three times relative to the cold process.

As well as the penetration depth, the dilution is an important quality indicator for the DED process [40,41]. If the value is too low, there is a high probability of insufficient melting of the previous layer accompanied with lack of penetration and LoF. However, if too high, excessive remelting can lead to impaired microstructure and mechanical properties. In the case of the performed experiments, all the specimens characterized by a dilution value above 32% proved to have sufficient melting of the substrate to avoid LoF defects. In the stable process conditions, the beads of the best quality were characterized by the dilution

at the level between 41% and 44%. This corresponds to the experiments with the applied hot-wire level L3 and the medium laser power as well as the hot-wire level L2 and the high laser power.

The results of the statistical analysis confirm that both factors have significant influence on the dilution. Still, similarly to height and width, the laser power is the most influential one and the interaction of the two factors does not have an influence on the parameter, see the Pareto charts in Figure 10d.

### 3.4.5. Internal Defects and Surface Condition

The visual inspection revealed a difference in the surface profile of the beads depending on the process conditions. The beads created when the material transfer was smooth and the process was stable were characterized by the smooth surface, see Figure 11b image to the left. Whereas, when depositing in the unstable process conditions, the resulting profile of the surface was rough with clearly distinguishable grooves, see Figure 11b image to the right. The grooves were created when the melt pool was disturbed during the occurrence of an arcing or a droplet detachment described in Section 3.2. This type of uneven surface creates favorable conditions for crack initiation as they can serve as local stress concentration areas when material is subjected to stresses related to loads and deformations [42]. It also increases the potential of LoF or pore occurrence when depositing subsequent layers. This is due to the created variations of distance between the surface of the previous layer and the wire tip. Depending on the extent and the direction of the distance change, droplets or stubbing can occur [23] which promote creation of the internal defects.

Although observed in the camera images, no indications of spatter were noticed on the surfaces of the beads and the substrates.

Among the three common types of defects for LDEDw, i.e., cracks, pores and LoFs, only the latter ones were found in the deposits, see Figure 11a. The LoFs occurred at the locations in close vicinity to the middle of a bead, where the wire entered the melt pool. At these locations, a clear boundary between a solid substrate and melted wire was recognizable. The defects were created in cold process conditions for the low and medium laser power, which is in agreement with the observation of the wire shadowing the substrate from the laser beam, see Section 3.2. Thus, the energy supplied by the laser beam was not high enough to melt both the wire and the substrate. The lengths of LoFs perpendicular to the bead direction varied between 0.01 mm and 0.46 mm. In general, the presence of LoF defects is highly detrimental for the material properties since they can act as stress concentrators and crack initiators [43]. Due to this, it should be avoided by any possible means. The conducted experiments show that the stable and LoF free process conditions can be achieved either by applying the high laser power or by the combination of electrical power in the range from 65 W to 190 W with the medium laser power. The second alternative allows more cost-efficient deposition, where the laser power is used more efficiently.

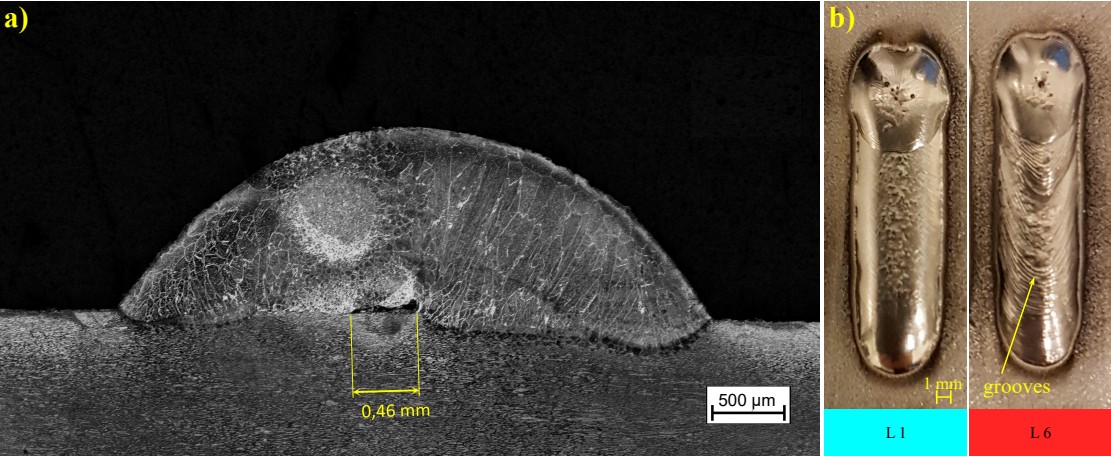

**Figure 11.** The examples of defects identified in the deposited beads: (**a**) the lack of fusion in the middle of the bead, (**b**) a comparison of surface conditions obtained by applying two different levels of the hot-wire–smooth surface, the hot-wire level L1 (image to the left), rough surface with groves, the hot-wire level L6 (image to the right).

## 4. Conclusions and Perspectives

This study investigated the process of Laser-Directed Energy Deposition of a Duplex stainless steel wire pre-heated by an electric current flowing through it. Hence, the energy required to melt the material was supplied by two independent sources: the laser beam and the electrical current. The impact of six levels of resistive pre-heating of the wire on the stability of the deposition process as well as the morphology of the deposited beads was evaluated under the three levels of laser power. Based on the findings the following conclusions have been drawn.

Supplying power from two independent sources influences the characteristics of the process, creating a new Hot-Wire Laser-Directed Energy Deposition process. The resistive pre-heating of the feedstock wire allowed more flexible adjustments of the process window by fine-tuning the heat input. This resulted in improved metal transfer and metal fusion in the stable process conditions.

The application of two independent energy sources provided new means of controlling the beads' geometry. By increasing the resistive pre-heating, it was possible to enhance the fusion, thus the penetration depth, avoiding the increase in bead width accompanied by a decrease in bead height, which are common byproducts when increasing only the laser power.

The overall efficiency of the material heating was improved by the application of the resistive pre-heating. Not only the efficiency of the electrical power was high, close to 100%, but also the pre-heating of the feedstock wire increased its absorptivity to the laser light. The experiments showed that the improved process stability and the metal fusion was achieved by adding about 15 times less Watts of generated electrical power in comparison to laser power. Hence, adding even a few dozen Watts of electrical power can prove to be a viable option for improving the deposition process and preventing the presence of detrimental LoF defects.

The improved efficiency of heating the wire as well as improved quality of the beads while processing with HW-LDED indicate a potential for future applications of hybrid systems where electrical power is combined with laser power. For such systems, the more efficient usage of the laser energy would enable to decrease the necessary laser power to create sound process conditions. Thus, applications of lower power laser sources could be possible which would reduce the overall costs of equipment. Hence, the HW-LDED has the potential to become a viable alternative to WAAM in large-scale production, while keeping the advantages of laser-based technologies. Although this work presents only the process results for Duplex stainless steel, similar influence of the addition of the hot-wire can be expected for other metallic alloys processed with LDEDw.

The voltage and the current supplied to the wire can be used not only for pre-heating but also as an indication of the process stability. During the experiments, the increased variation in electrical signals as well as the shift of signal levels indicate process disturbances. Thus, the hot-wire system can be used for in-process monitoring. If further developed, this technology can be a reliable, low-cost solution for an industrial application as the system can easily be integrated into an existing LDEDw hardware and is robust enough to withstand harsh industrial environments.

**Author Contributions:** Conceptualization, A.K. and P.H.; methodology, A.K. and P.H.; validation, D.S.; formal analysis, A.K. and K.T.P.; investigation, A.K., P.H. and K.T.P.; writing—original draft preparation, A.K.; writing—review and editing, K.T.P., M.A.V.B., F.S., P.H., D.S. and A.A.; supervision, F.S. and A.A.; project administration, F.S.; funding acquisition, F.S. and A.A. All authors have read and agreed to the published version of the manuscript.

**Funding:** This work was supported by the Swedish Knowledge Foundation within the project SUMANnext [20160281] and the project SAMw [20170060] as well as Sweden's innovation agency Vinnova within the project InAIRwire [2019-02752].

**Institutional Review Board Statement:** Not applicable.

**Informed Consent Statement:** Not applicable.

**Data Availability Statement:** The data presented in this study are available on request from the corresponding author. The data are not publicly available due to temporary lack of access to a public repository.

**Acknowledgments:** The authors would like to thank Morgan Nilsen and Yongcui Mi for their help with setting up the camera monitoring system and image data acquisition while running experiments, Anna-Karin Christiansson for her support and valuable advice regarding the experimental work and writing assistance as well as Isabelle Choquet at for her valuable guidance in topic of welding and additive manufacturing. All mentioned colleagues are affiliated with University West in Trollhättan, Sweden.

**Conflicts of Interest:** The authors declare no conflict of interest. The funders had no role in the design of the study; in the collection, analyses, or interpretation of data; in the writing of the manuscript, or in the decision to publish the results.

## Abbreviations

The following abbreviations are used in this manuscript:

| | |
|---|---|
| DED | Directed Energy Deposition |
| AM | Additive Manufacturing |
| LoF | Lack of Fusion |
| LDEDw | Laser-Directed Energy Deposition with wire |
| LMDw | Laser–Metal Deposition with wire |
| WAAM | Wire-Arc Additive Manufacturing |
| EPS | Electrical Power Source |
| GTAW | Gas Tungsten Arc Welding |
| AC | Alternating Current |
| HWAAM | Hot-Wire-Arc Additive Manufacturing |
| LBW | Laser Beam Welding |
| HW-LDED | Hot-Wire Laser-Directed Energy Deposition |
| PLC | Programmable Logic Controller |
| DC | Direct Current |
| LED | Light Emitting Diode |
| CMOS | Complementary Metal Oxide Semiconductor |

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
