# Peer review of "Hot-Wire Laser-Directed Energy Deposition: Process Characteristics and Benefits of Resistive Pre-Heating of the Feedstock Wire"

_metals, doi:10.3390/met11040634_

Round 1

Reviewer 1 Report

The research is of great engineering significance and the presentation is reasonably good, but the paper can be improved by addressing the following issues: 

(1) The grammatical errors should be corrected. Here are some examples only: In line 13, "mean" should be changed to "means". In line 25, "have begun" should be changed to "has begun".  

(2) Indentation should be added to the beginning of each paragraph, or else it would be difficult to differentiate different paragraphs. So many long paragraphs make the paper difficult to read. 

(3) The duplex steel used is an austenite + ferrite duplex stainless steel. Some metallographs at higher magnifications should be included to show the microstructures. Especially, It would be good to study and state how the resistive heating affects the percentage of austenite and ferrite in the deposited material.  

Author Response

Manuscript ID: metals-1156781

Title: Hot-Wire Laser Directed Energy Deposition: Process Characteristics and Benefits of Resistive Pre-Heating of the Feedstock Wire

Author: Agnieszka Kisielewicz et al.

Dear Reviewers,

We would like to thank all of you for your time and useful comments allowing us to improve our manuscript. The manuscript has been revised, taking into account all the suggestions. All major changes to the manuscript are highlighted in yellow. Please find detailed responses to all your comments in the attached file.

I hope the paper may now be accepted for publication in Metals: Special Issue "Advances in Metal Additive Manufacturing: New Materials, Process Enhancement, Monitoring and Sustainability".

Yours sincerely,

Agnieszka Kisielewicz.

_____________________________________________________________________________________

The authors would like to inform all the reviewers that, except the suggested improvements by the reviewers, there have been a few additional changes introduced to the manuscript.

It was noticed that the stated frame rate of high-speed camera was incorrect. The mistake was corrected, see page 7, lines 199-200.

Minor grammar improvements were not marked in the manuscript, yet if a sentence required major improvement its new version was highlighted in yellow.

Reviewer 2 Report

The submission was carefully reviewed and was recommended for publication. However, some points needed to be explained or modified before any proceeds. The provided remarks are as follows and the authors are suggested to consider the comments while revising their manuscript:
(1) The language of the paper needs a thorough polishing. Many sentences contain language mistakes. Some sentences on the first page are listed below:
--feedstock wire (here 2 called hot-wire) on stability of laser directed => feedstock wire (here 2 called hot-wire) on the stability of laser directed 
--were used to reveal 4 process characteristic and its stability window. => were used to reveal 4 process characteristics and their stability window. 
--This resulted in a better process stability as well => This resulted in better process stability as well
--In case of the applications using wire, => In the case of the applications using wire,
--Hence, the applications of wire-feed 28 DED can be found in aerospace industry, => Hence, the applications of wire-feed 28 DED can be found in the aerospace industry,
+As you can see, many sentences contain language mistakes on only the first page of the manuscript. Careful polishing is required before publication as some sentences are hard to understand or may misleading to the reviewer/reader.
(2) The number of the reviewed refs is limited and also the ref part contains many outdated refs seek substitution with recent publications. 
(3) In Table2 the different levels of the voltage for hot-wire are mentioned; however, the voltage is not highly meaningful as it could have a complex contribution in heating-up the wire when different systems are applied for warm-up. It is recommended to add the given wire temperature by each voltage level here for better understanding. 
(4) Fig. 7 is not meaningful enough. What is the time frame for the images? What one may understand from these micrographs?
(5) As claimed by the authors, the discussion part on the possible defect formation must be extended within the revised manuscript. 

Author Response

(The authors gave the same response as above.)

Reviewer 3 Report

Dear Authors,

I have read your paper "Hot-Wire Laser Directed Energy Deposition: Process Characteristics and Benefits of Resistive Pre-Heating of the Feedstock Wire" carefully. 

This paper describes the effect of resistive heating of the wire on the LDED process. 

The paper is easy to read.

But the methods are not properly described, so that other research groups may not reproduce them.

The paper is interesting. However, it requires few corrections.

  1. Please, add the chemical composition of the wire and substrate. 
  2. Figure 6, 7, 11. Please, add the dimension marker. 
  3. Please specifically discuss the advantages of your work. Some parts of your conclusion can be writing in the discussion section. Is there the effect of the diameter of the wire on the process? 

 The paper can be accepted for publication only after major improvements.

Author Response

(The authors gave the same response as above.)

Round 2

Reviewer 3 Report

Dear Authors,

I have read your modified paper "Hot-Wire Laser Directed Energy Deposition: Process Characteristics and Benefits of Resistive Pre-Heating of the Feedstock Wire" carefully.

Explanations are clear and the paper is easy to read.

I can recommend the Editor to accept this revised manuscript to be published in Metals.